# Safeguarding Bee Health: Insights from a Collaborative Monitoring and Prevention Project Against Pesticide Poisonings

**DOI:** 10.3390/ani15030449

**Published:** 2025-02-06

**Authors:** Mara Gasparini, Giovanni Prestini, Franco Rainini, Gabriella Cancemi, Silvia De Palo, Livio Colombari, Michele Mortarino

**Affiliations:** 1Chemistry Laboratory, Istituto Zooprofilattico Sperimentale della Lombardia e dell’Emilia, 25124 Brescia, Italy; mara.gasparini@izsler.it (M.G.);; 2Agenzia di Tutela della Salute della Brianza, 20900 Monza, Italy; giovanni.prestini@ats-brianza.it (G.P.); franco.rainini@tiscali.it (F.R.); 3Associazione Apicoltori Lombardi, 25124 Brescia, Italy; info@apicoltori.so.it; 4Associazione Produttori Apistici Apilombardia, 27058 Voghera, Italy; livio.colombari@apilombardia.it; 5Department of Veterinary Medicine and Animal Sciences, University of Milan, 26900 Lodi, Italy

**Keywords:** honey bees, pesticides, bee health, environmental pollution, sublethal effects, monitoring network, pesticide analysis, gas and liquid chromatography tandem quadrupole mass spectrometry, pyrethroids, neonicotinoids

## Abstract

In recent years, bee poisoning caused by pesticides and agrochemicals has become a growing problem, harming honey bee health and contributing to Colony Collapse Disorder (CCD). One of the key challenges in preventing this problem is the lack of sufficient knowledge about how much bee colonies are exposed to these substances. To tackle this, it is crucial for beekeepers, local authorities, and researchers to work together, sharing data and taking timely action to prevent poisoning events. This study highlights the results from a collaborative project in an area of northern Italy, where four sentinel apiaries were set up to monitor pesticide exposure. Pollen samples were collected and analyzed for pesticides, revealing the presence of harmful chemicals like pyrethroids, amitraz, and others. In one case, acute bee mortality was linked to pyrethroid exposure in an urban area. The findings stress the need for better monitoring and cooperation between beekeepers and authorities to protect bee populations and ensure sustainable beekeeping practices.

## 1. Introduction

In recent decades, the issues of lethal and especially sublethal bee poisonings have increased, due to pesticides and other substances used in agriculture for crop protection (insecticides, herbicides, fungicides), in urban environments (especially insecticides for mosquito control and other harmful insects), and in floriculture. The phenomenon of pesticide poisonings represents one of the emerging health issues in the beekeeping field and one of the main causes of the so-called Colony Collapse Disorder (CCD), along with parasitic diseases (such as varroosis caused by the mite *Varroa destructor*), damage to colonies caused by pests (like the small hive beetle *Aethina tumida*), phenomena related to climate change, and related nutritional stresses [1,2,3]. It is particularly important to differentiate between lethal and sublethal poisonings in bees due to pesticide exposure: in fact, sublethal effects can often be overlooked but have a significant impact on the health of bee colonies [4]. Furthermore, long-term effects of pesticide exposure on bee populations and bee communities can occur. Specific active principles such as neonicotinoids, organophosphates, and pyrethroids are considered to be more involved in these phenomena [5,6,7].

In the European Union, field data on this phenomenon are periodically requested through questionnaires to be administered to beekeepers or arise from the interventions of local health authorities in the territory following episodes of suspected bee poisoning. In some cases, evidence of lethal and sublethal poisoning has been supported by laboratory tests conducted on various hive matrices and has led over the years to the prohibition or limitation of the use of some pesticides (neonicotinoid insecticides and others) that have been recognized as highly toxic to bees [8]. Thus, there is still an urgent need to develop local programs aimed at monitoring and preventing bee poisonings. In this regard, some potential issues need to be overcome: (a) delays in reporting by beekeepers, (b) difficulties in sampling bees and other matrices (pollen–honey) due to untimely intervention and the lack of specific sampling equipment, (c) insufficient knowledge of sampling methods and of the correct preservation and transportation of samples, (d) insufficient knowledge of the environmental dynamics of many pesticides, thus making it difficult to distinguish between widespread pollution and single episodes of point contamination, and (e) reduced awareness of the potential exposure of bee colonies to residues of acaricide treatments that may persist long-term in hive matrices, in addition to contaminants of an environmental source.

In order to help address at least some of the issues described above, collaboration is needed among various stakeholders, including beekeeping associations, local veterinary authorities, and researchers, to develop monitoring programs and implement prevention measures for potential bee colony poisoning events caused by such contaminants.

With this aim, the present study reports the results of a monitoring and surveillance activity performed by a network of collaboration among beekeepers, researchers, and local health authorities in an area of northern Italy.

## 2. Materials and Methods

### 2.1. Sentinel Apiaries

In 2021, four sentinel apiaries (hereafter referred to as S1 to S4), each composed of three beehives equipped for pollen collection and dead bee sampling, were placed in the provinces of Monza and Brianza (MB) and Lecco (LC), Lombardy region, northern Italy, in locations that differed from each other based on possible sources of pollution, as described below. In detail, the sentinel apiaries were located in the following municipalities: S1 in Agrate (MB); S2 in Monza (MB); S3 in Valmadrera (LC); and S4 in Crandola Valsassina (LC) (Figure 1).

S1 and S2 were located in flat areas characterized by urban–industrial settlements, with high-traffic roads. In particular, the area within the potential bee flight range around S1 was characterized by the highest vehicular traffic density among all the sampling sites, being near the intersection of two highways traveled daily by tens of thousands of vehicles. In addition, the S1 area was characterized by the presence of floriculture companies and by a remarkably higher intensity of agricultural production (mainly cereal, soybean, and other oilseed crops) compared to S2; conversely, the area within the flying range of the bees from S2 was characterized by a higher population density and presence of parks, public gardens, and tree-lined avenues compared to S1.

S3 was located at 450 m above sea level, with the potential bee flight range covering a hilly area of an altitude from 230 to about 1400 m above sea level, with the presence of uncultivated pastures and spontaneous flora and a waste incineration plant. In comparison to the areas where S1 and S2 were located, the area within the potential bee flight range around S3 showed a lower presence of urban–industrial settlements and high-traffic roads.

S4 was located at 850 m above sea level, with the potential bee flight range covering a mountainous area of an altitude from 600 to about 2000 m above sea level. The S4 area was mainly characterized by uncultivated pastures and spontaneous flora, with the lowest presence of urban–industrial settlements and high-traffic roads among all the sampling sites.

### 2.2. Sampling Activities

The beehives of S1–S4 were equipped with pollen traps with a removable drawer closure and underbasket traps for collecting dead bees. From April to September in both 2021 and 2022, pollen samples were collected once or twice a month depending on the pollen collection activity of honey bees and on major agronomic activities involving pesticide and agrochemical use. During each collection, samples were prepared by combining pollen from the three hives of each sentinel apiary. This approach ensured the collection of sufficient pollen quantities during the entire sampling period, even at times when the harvest was scarce. A total of 20 pollen samples were collected for each sentinel apiary during the whole study.

### 2.3. Chemical Analysis

The pollen samples collected during the monitoring activity were refrigerated during transportation and delivered, after freezing them upon arrival, to the Chemistry Laboratory of the Istituto Zooprofilattico Sperimentale della Lombardia e dell’Emilia Romagna, Brescia, Italy.

The pollen samples were analyzed for the detection of the target molecules of pesticides and agrochemicals listed in Appendix A. The limit of quantification for all analytes was 10 ppb (μg/kg), except for Iprodione (20 ppb). The analysis for the detection of pesticides and phytopharmaceuticals was carried out applying the QuEChERS technique for the extraction and purification of pollen samples, followed by gas and liquid chromatography tandem quadrupole mass spectrometry (GC-MS/MS and LC-MS/MS). Briefly, 5 g of a homogenized pollen sample was weighed, then a volume of 10 mL of deionized water was added and the mixture was extracted with 10 mL of acetonitrile using QuEChERS (Agilent Technologies, Santa Clara, CA, USA) extraction salts, containing 1 g of sodium chloride, 4 g of magnesium sulfate, 1 g of sodium citrate, and 0.5 g of sodium hydrogen citrate sesquihydrate. After centrifugation, 6 mL of the extract was cleaned up in a dispersive solid-phase extraction tube (dSPE) containing 150 mg of PSA and 900 mg of magnesium sulfate (Agilent Technologies, Santa Clara, CA, USA); no subsequent sample extract concentration step was necessary, nor was a solvent change for instrumental analysis. Each pollen sample’s purified extract was directly analyzed using GC-MS/MS and LC-MS/MS techniques, as required for the detection of the pesticides with different chemical and physical properties listed in Appendix A. GC-MS/MS instrumental analysis was conducted with a GC system (model 7890B, Agilent Technologies, Santa Clara, CA, USA) coupled with a triple-quad mass spectrometer (model 7010, Agilent Technologies, Santa Clara, CA, USA). The chromatographic separation was obtained using a fused silica capillary column (30 m, 0.25 mm, 0.25 µm), the ZB-MultiResidue-1 (Phenomenex, Torrance, CA, USA). Helium was used as the carrier gas at 1.1 mL min^−1^. A volume of 0.2 µL of the purified pollen sample extract was injected in splitless mode at a temperature of 230 °C. The GC oven’s initial temperature was 70 °C, held for 2 min, then ramped at 25 °C min^−1^ to 150 °C, then ramped at 12 °C mni^−1^ to 200 °C, then ramped at 6 °C min^−1^ to 310 °C, held for 7 min, for a total run time of 34.7 min. The MSD transfer line temperature was 310 °C. The mass spectrometer source operated in EI mode at 70 eV at a temperature of 280 °C. LC-MS/MS instrumental analysis was conducted with an LC system (model 1290 Infinity HPLC, Agilent Technologies, Santa Clara, CA, USA) coupled with a triple-quad mass spectrometer (model 6470, Agilent Technologies, Santa Clara, CA, USA). The chromatographic separation was obtained using an Acquity UPLC HSS T3 column with dimensions of 10 cm, 2.1 mm, 1.8 µm (Waters, Milford, MA, USA). The mobile phases were 2 mM ammonium formate and a 2 mM formic acid–water solution containing 2% methanol (mobile phase A) and 2 mM ammonium formate and a 2 mM formic acid–methanol solution (mobile phase B). A volume of 1 µL of the pollen sample’s purified extract was injected. The flow rate was kept constant at 0.3 mL min^−1^, and the gradient program was set as follows: 95% A was the initial condition for 1 min, followed by a linear gradient down to 55% over 2 min and then down to 5% in 13 min, after which the mobile phase composition was maintained at 5% for 2.9 min, then the re-equilibration time was 4 min, for a total run time of 25 min. The mass spectrometer source operated in ESI-positive mode.

For both GC-MS/MS and LC-MS/MS analysis, the triple-quadrupole systems worked in Multiple Reaction Monitoring (MRM) scan mode. For each analyte, two transitions were monitored. To prepare matrix-matched calibration curves for the pesticide quantification, a blank pollen sample, where no pesticide listed in Appendix A was detected in a previous analysis, was used. Several standard solutions in a concentration range of 100–250 mg L^−1^, containing all the molecules analyzed, were purchased from CPAchem LTD (Bogomilovo, Bulgaria). These solutions were mixed to obtain the standard mix working solutions used to prepare the matrix-matched calibration curves used to identify and quantify pesticides in the pollen samples. This method was validated according to EU official protocols [10], enabling the assessment of potential correlations among the analytes. The pesticide accumulation in pollen and all samples was analyzed in accordance with the criteria described in [10].

## 3. Results

Regarding the analysis of pollen samples, 5 pesticide molecules out of a total 238 searched compounds were detected, with a rate of 1.68% in 2021 and 2.1% in 2022, respectively, and quantities ranging from 0.01 to 0.16 ppm. The herbicide Pendymetalin was detected in samples collected during May 2021 from S1, S3, and S4. During 2022, three out of only four Pendymetalin-positive samples were from S1, while only one positive sample was from S3. Overall, most positive samples were collected during April and May. Remarkably, all samples collected from S2 during the whole study were negative (Table 1).

Regarding the pyrethroid insecticides, permethrin and cypermethrin were only detected in pollen samples from S2, as follows: (a) 3 out of 10 samples (30%) were collected during the period from June to early July 2021; (b) 9 out of 10 samples (90%) were collected throughout the monitoring season from April to September 2022, while the only negative sample was collected in the first half of May (Table 2).

During the observation period, no acute episodes of adult bee mortality or depopulation were detected in all the sentinel apiaries under monitoring. Notably, in mid-June 2022, the local Veterinary Health authority was notified of a mortality event in Monza (MB) municipality, at a site located at a straight-line distance of approximately 700 m from S2. At this site, dead or dying bees and bumblebees with neurological symptoms were found under flowering common lime trees located in residential areas with parks and gardens with pools. The mortality lasted for approximately ten days, during which samples of bees and bumblebees were collected and subjected to pesticide analysis, with positive results for the pyrethroids permethrin and cypermethrin at a concentration of up to 1.1 ppm, i.e., over five times the LD_50_. Remarkably, the same pesticides were found in 9 out of 10 pollen samples (90%) collected at S2 throughout the monitoring period from April to September 2022 (Table 2). During this lethal event, the observation and counting of dead bees were also conducted at S2, with no significant abnormal mortality observed.

Amitraz and its degradation products were detected in 6 out of 40 samples and in three out of four sentinel apiaries (excluding S4) in 2021 and in 7 out of 40 samples across all sentinel apiaries in 2022 (Table 3).

Coumaphos was only detected in four out of ten samples collected from S2 during 2022, with a concentration ranging between 0.012 and 0.130 ppm. It should be noted that Coumaphos is currently out of legal use as a varroacide in the EU. Then, in order to investigate any possible illegal treatments by a beekeeper, a supplementary inspection was conducted at S1, and information was collected on possible changes in hive management between 2021 and 2022. In this regard, it was reported by the beekeepers that in 2022 the monitored bee colonies were transferred to old wooden hives that had been stored in the warehouse since 2001 after a long time of use in the field. Thus, samples of wood, wax, and propolis were collected by scraping from the internal walls of these hives and sent to the Istituto Zooprofilattico Sperimentale delle Venezie, Legnaro, Padua, Italy, for pesticide analysis. As a result, only Coumaphos was detected at up to 0.037 ppm, whereas no other active compounds were absent.

Regarding neonicotinoids, they were not detected in any of the pollen samples collected during the whole study.

## 4. Discussion

There is a growing spread and persistence of many pesticides (or their metabolites) in various plant, animal (including bees, pollen, and sometimes honey), and inorganic matrices, even at relevant temporal and spatial distances from the sites and periods of use [11,12]. Several studies showed that, even at low concentrations, the presence of insecticides in various beehive matrices can significantly damage colony development [13]. This primarily concerns pollen, which is used for larval feeding by bee nurses, and its various sublethal and chronic pathogenic effects. These may lead to a significant decrease in the queen’s fertility, as well as a reduction in bees’ longevity, immune function, orientation, and social behavior. Moreover, these effects may contribute to CCD, often in combination with other factors [14]. Further research also demonstrated the risk of severe acute, sublethal, and chronic poisoning not only for honey bees but also for several wild pollinators, associated with the decline in agricultural food production linked to entomophilous pollination [15].

In this study, searching for pesticides in the pollen samples using analytical protocols for non-polar molecules was justified since many pesticides already known to be potentially toxic for honey bees are non-polar and can persist for a long time in the environment [16,17]. During the two years of the present monitoring study, pesticides were found in pollen at relatively low concentrations and, in any case, were not sufficient to cause observable acute episodes of the mortality or depopulation of adult bees in the sentinel apiaries. The comparison of the results obtained in the year 2022 with those of 2021 highlights an increased detection of pyrethroids in one sentinel apiary, whose neighboring area was affected by an episode of acute bee mortality, probably due to poisoning due to the same pesticides. Therefore, it is conceivable that bee colonies provided larvae with pollen contaminated with pyrethroid insecticides, with possible sublethal chronic effects that, despite being barely detectable, are already described in the literature [18]. Notably, permethrin and cypermethrin are currently banned in agriculture, while they are contained in various commercial products authorized for professional and/or domestic use against harmful insects, mainly adult mosquitoes [19]. Therefore, it could be hypothesized that in the area surrounding S2, adulticide treatments against mosquitoes were performed without complying with regulatory provisions designed to protect bees and other beneficial insects from the off-target effects of such treatments [20].

The herbicide Pendymetalin was also detected, although not regularly across the different study areas and during the monitoring period. This active compound is contained in some commercial products regularly authorized in Italy and used almost exclusively in both professional and private agriculture, primarily as a pre-emergence herbicide on spring crops [21].

Amitraz and Coumaphos are varroacides, and the significance of their detection during the present study must be discussed separately from agrochemicals such as pendimethalin and from pyrethroids primarily intended for use in urban settlements. Regarding amitraz, the intact molecule and its degradation products were found in both spring and autumn 2021, while in 2022, they were only found in late summer/autumn. This presence was likely linked to treatments against varroosis in bees, as amitraz is the active principle of licensed miticides available as impregnated plastic strips to be inserted into hives and was also used in the hives of the sentinel apiary sites in previous seasons and/or during the study period [22]. The persistence of amitraz and its degradation products well beyond the treatment period, usually performed during late autumn–early winter, was supported by positive detection in spring 2021. Remarkably, it is known that the long-term persistence of amitraz and its degradation products could lead to the onset of resistant varroa mites with decreased treatment efficacy [23].

Coumaphos was detected only in S1. It should be noted that before 2001, licensed miticides containing this active compound were widely used in beekeeping for varroa mite control. Notably, it is reported that Coumaphos residues may persist in the wax and other non-polar matrices of the hive for several years after acaricidal treatment [23,24]. On this basis, it is possible to hypothesize that the pollen contamination detected in S1 hives was linked to Coumaphos residues still present after many years of storage in the old wooden hives to which the S1 colonies were transferred and not from any illegal, recent use of this pesticide by the beekeeper.

It is noteworthy that neonicotinoids were not detected in any of the pollen samples collected from the monitored beehives. This outcome suggests either limited or no exposure to these pesticides from the surrounding environment or effective management practices that minimize their presence in bee-attractive plants. Such findings are encouraging, as neonicotinoids have been implicated in bee health issues, and their absence underscores the importance of continued monitoring and potentially implementing strategies to mitigate pesticide exposure in bee habitats [25].

## 5. Conclusions

Overall, this study shows that bee colonies can be exposed to contamination based on the different characteristics of the territory in which they are located and supports the importance of integrated initiatives for the surveillance and prevention of poisoning events at the territorial level. The results confirmed that non-polar pesticides and phytopharmaceuticals potentially toxic to bees may be detected in pollen samples from sentinel apiaries located in areas of different geographic characteristics, urban densities, and agricultural intensities.

Furthermore, the findings of widespread contamination by acaricides like amitraz currently under use in beekeeping, together with the episodical detection of long-persisting, unrenewed molecules such as Coumaphos, make it appropriate to implement future programs for monitoring resistance to acaricides.

For these purposes, the establishment of collaboration networks between public institutions and other stakeholders, such as the one described in this study, can allow for in-depth field investigations in areas with different pollution risks. Furthermore, it is possible to establish connections between acute bee poisoning episodes and the detection of widespread contamination in the same areas.

## Figures and Tables

**Figure 1 animals-15-00449-f001:**
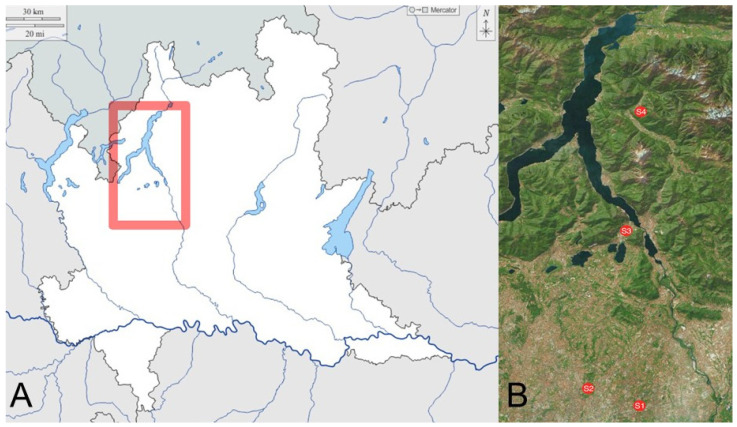
Representation of the study area, delimited in red on the map of the Lombardy region (panel (**A**)), and the details of the location of the sentinel apiaries S1–S4 (panel (**B**)). Figure adapted from the original available in [9].

**Table 1 animals-15-00449-t001:** Pendymetalin detection in pollen samples. Values are expressed in ppm (mg/kg).

Year	Sentinel Apiary	April	May/1	May/2	June/1	June/2	July	August/1	August/2	September/1	September/2
2021	S1	0	0.016	0	0	0	0	0	0	0	0
	S2	0	0	0	0	0	0	0	0	0	0
	S3	0	0.015	0	0	0	0	0	0	0	0
	S4	0	0.012	0	0	0	0	0	0	0	0
2022	S1	0.015	0.014	0	0	0	0	0.012	0	0	0
	S2	0	0	0	0	0	0	0	0	0	0
	S3	0	0.012	0	0	0	0	0	0	0	0
	S4	0	0	0	0	0	0	0	0	0	0

**Table 2 animals-15-00449-t002:** Permethrin and cypermethrin detection in pollen samples from S2. Values are expressed in ppm (mg/kg).

Year	Target	April	May/1	May/2	June/1	June/2	July	August/1	August/2	September/1	September/2
2021	Cypermethrin	0	0	0	0.110	0.021	0.031	0	0	0	0
	Permethrin	0	0	0	0	0.017	0	0	0	0	0
2022	Cypermethrin	0.013	0	0.014	0.012	0.015	0.160	0.110	0.050	0.140	0.032
	Permethrin	0.011	0	0.017	0.018	0.017	0	0	0	0	0

**Table 3 animals-15-00449-t003:** Amitraz detection in pollen samples. Values refer to the total content of amitraz and its degradation products, expressed in ppm (mg/kg).

Year	Sentinel Apiary	April/1	April/2	May/1	May/2	June/1	June/2	July	August	September/1	September/2
2021	S1	0	0	0	0	0	0	0	0	0.021	0.018
	S2	0.029	0.022	0	0	0	0	0	0	0	0
	S3	0.029	0	0	0	0	0	0	0	0	0
	S4	0	0	0	0	0	0	0	0	0.032	0
2022	S1	0	0	0	0	0	0	0	0	0	0.022
	S2	0	0	0	0	0	0	0	0	0	0.024
	S3	0	0	0	0	0	0	0.046	0.088	0.180	0.130
	S4	0	0	0	0	0	0	0	0.020	0	0

## Data Availability

The data are available upon request to the authors.

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
