# Peer review of "Safeguarding Bee Health: Insights from a Collaborative Monitoring and Prevention Project Against Pesticide Poisonings"

_animals, 2025, doi:10.3390/ani15030449_

Round 1

Reviewer 1 Report

Comments and Suggestions for Authors

Thank you for this interesting paper, which provides deeper insight into the health risks coming from pesticide applications. However, there are some terms in the paper that aren't correct or not properly introduced. Consider that the terms in English may not be a direct translation from the Italian term ("Hive Depopulation Syndrome") and check the papers you cite for the correct terminology. Or, if you think that your wording provides an additional element of understanding, define the terms you use. I began commenting on this, but stopped after the introduction to focus more on the "flesh" of the paper. 

Line 20: honeybee numbers not "honeybees' numbers". "Hive depopulation syndrome" sounds more Italian than English to me. It's usually referred to as CCD or Colony Collapse Disorder. 

Line 22: honeybee health, like above. Check this throughout the paper.

Line 37: in AN urban area.

Line 47: I understand what you mean by clinical and subclinical, but as these terms aren't the usual way of describing the phenomenon, you may add a short definition of what you mean. 

Line 54: "hive depopulation syndrome" isn't the right term in English, see above and change throughout the paper.

Line 55: Put species names in italics. Apis mellifera etc. Please change throughout the paper. In addition, Aethina tumida isn't at the same level like Varroa destructor. And both aren't really diseases, varroa is a parasite causing a disease called varroosis and the Small Hive Beetle is more a pest, not a pathogen. 

General question: why are you using clinical and subclinical? Why don't you stay with the usual terms? Like acute poisoning, sublethal effects, etc? Again, this needs a definition if you have a reason for using different terms.

Line 65: European Union

Line 95: better "sampling" instead of retrieval.

Line 213: why is the number in a different font? Are the numbers checked?

Lines 215-217: This reads like the indications from the template, please delete. 

Lines 247 ff: Amitraz and Coumaphos are varroacides. Both substances are not used in agriculture in the EU. So, it would be good to classify this substances accordingly. Coumaphos should be out of use as a varroacide (resistance issues) but is still in use in DIY treatments by beekeepers. So, in case of any incidents with this substance, it's not agriculture to blame, but illegal treatments from beekeepers. 

Line 288: As noted above, you can't classify amitraz and coumaphos the same way you do with the agricultural pesticides pendimethalin and the detected pyrethroids.

The discussion overall is very unstructured and not totally backed up by your data. This need a more concise and streamlined discussion, especially going into the fact of the different sources of pesticides (agriculture and veterinarian). You do this in the end, but the lack of structure makes it difficult to follow. This needs major revision. It reads more like a draft.

Author Response

We present here a point-by-point response to Reviewers’ comments on the manuscript: “Safeguarding Bee Health: Insights from a Collaborative Monitoring and Prevention Project against Pesticide Poisonings”, submitted to be published on Animals (animals-3316322). The relevant revisions are highlighted in yellow in the text of the manuscript.

Reviewer #1: Thank you for this interesting paper, which provides deeper insight into the health risks coming from pesticide applications. However, there are some terms in the paper that aren't correct or not properly introduced. Consider that the terms in English may not be a direct translation from the Italian term ("Hive Depopulation Syndrome") and check the papers you cite for the correct terminology. Or, if you think that your wording provides an additional element of understanding, define the terms you use. I began commenting on this, but stopped after the introduction to focus more on the "flesh" of the paper.

We wish to thank the reviewer for his/her constructive comments and his/her ideas to improve the manuscript.

The main detailed comments:

Line 20. honeybee numbers not "honeybees' numbers".

Done.

"Hive depopulation syndrome" sounds more Italian than English to me. It's usually referred to as CCD or Colony Collapse Disorder.

Throughout the whole text, the original expression "Hive depopulation syndrome" was replaced with “Colony Collapse Disorder-(CCD)” as a more accepted and scientifically sounding term (e.g., Line 18)

Line 22. honeybee health, like above. Check this throughout the paper.

We checked and corrected the whole text as suggested (e.g., Line 18).

Line 37. in AN urban area.

Done as suggested.

Line 47. I understand what you mean by clinical and subclinical, but as these terms aren't the usual way of describing the phenomenon, you may add a short definition of what you mean.

As also suggested by Reviewer #2, we used the widely accepted terminology of lethal and sublethal throughout the manuscript.

Line 54. “hive depopulation syndrome” isn’t the right term in English, see above and change throughout the paper.

The term “hive depopulation syndrome” was replaced with “CCD” throughout the manuscript, as suggested.

Line 55. Put species names in italics. Apis mellifera etc. Please change throughout the paper.

Done.

Line 55. In addition, Aethina tumida isn't at the same level like Varroa destructor. And both aren't really diseases, varroa is a parasite causing a disease called varroosis and the Small Hive Beetle is more a pest, not a pathogen.

As suggested, we distinguished between parasitic diseases (such as varooosis caused by the mite Varroa destructor) and damages to colonies caused by pests (like the small hive beetle Aethina tumida).

General question: why are you using clinical and subclinical? Why don't you stay with the usual terms? Like acute poisoning, sublethal effects, etc? Again, this needs a definition if you have a reason for using different terms.

As already mentioned above, we used the widely accepted terminology of lethal and sublethal throughout the manuscript.

Line 65. European Union.

Done as suggested.

Line 95. European Union.

Since there were no references to the European Union in line 95 of the unrevised manuscript, we think this suggestion still applies to line 65 as above.

Line 95: better "sampling" instead of retrieval.

Done as suggested.

Line 213: why is the number in a different font? Are the numbers checked?

Checked as suggested.

Lines 215-217: This reads like the indications from the template, please delete.

Done. We sincerely apologize for having overlooked removing these instructions.

Lines 247 ff: Amitraz and Coumaphos are varroacides. Both substances are not used in agriculture in the EU. So, it would be good to classify this substances accordingly. Coumaphos should be out of use as a varroacide (resistance issues) but is still in use in DIY treatments by beekeepers. So, in case of any incidents with this substance, it's not agriculture to blame, but illegal treatments from beekeepers.

In Discussion (Lines 293-299), we explained more clearly our hypothesis that the pollen contamination detected in S1 hives was possibly linked to Coumaphos residues still present after many years of storage in the old wooden hives where the S1 colonies were transferred, and not from any illegal, recent use of this pesticide by the beekeeper. This explanation was supported by the results obtained from samples of hive matrices (wood, wax, and propolis) collected by scraping from the walls of these hives: only Coumaphos was found, that is well known as long-lasting in several non-polar matrices (we added a couple of supporting literature references).

Line 288: As noted above, you can't classify amitraz and coumaphos the same way you do with the agricultural pesticides pendimethalin and the detected pyrethroids.

In Discussion (Lines 280-282), we classified Amitraz and Coumaphos as varroacides, distinguishing them from agrochemicals such as Pendimethalin, and from pyrethroids primarily intended for use in urban settlements.

The discussion overall is very unstructured and not totally backed up by your data. This need a more concise and streamlined discussion, especially going into the fact of the different sources of pesticides (agriculture and veterinarian). You do this in the end, but the lack of structure makes it difficult to follow. This needs major revision. It reads more like a draft.

The discussion has been reformulated in accordance with the suggestion (Lines 245-300).

Reviewer 2 Report

Comments and Suggestions for Authors

The authors present a study on pesticide analysis in pollen. The study sound interesting. However, from reading the manuscript it becomes obvious that the manuscript was submitted unfinished. There are many technical errors. In addition, I feel that the manuscript was not originally written in English and that the translator was not of scientific background. It should not be published in the present form. The manuscript even has instruction from guidelines left within the text. It requires substantial revisions if it is to be reconsidered for publication.

Some specific comments:

L20: More accepted term is “colony collapse disorder” CCD.

L33: and is not needed?

L34: Replace “advanced chromatography techniques” with specific method(s) used.

L37: If it happened only once, particularly is not needed.

L38: clinical and subclinical? Do you mean lethal and sublethal? Please use widely accepted terminology of lethal and sublethal throughout the manuscript.  

L81-89: Specifics on the funding of the project go to the acknowledgment or funding section, not in the scientific part of the manuscript. Please describe the study not the project. This section should be the hypostases and/or aim of the presented study.

2.1 Monitoring stations

This whole section must be completely rewritten. Did you mean monitoring locations? What are sentinel stations? What are Plain Stations? Reading the description, it feels that these stations have the same characteristics. Station 1 designated as agricultural has significant industrial settlements and traffic. Then the same is for station 2. What does “Significant provincial, state, and municipal infrastructure” refer to?

L104: others? Please finish the sentence

L142: What is extraordinary monitoring?

A map of the locations would prove very useful.

L147: Exact model to be specified?

L148-151: From this sentence it can be concluded that the mortality was not measured at the sampling station but at hives adjacent to the station?

L158: Subtitle should be in a new line.

L166-176: How much of the supernatant after extraction procedure did you use for clean-up? With how much of the clean extract did you end up with? Did you have a concentration step? Did you follow literature protocol? If yes, please cite the work.

L207-209: How did you achieve matrix matched solutions?

L215-217: It is biggening to feel that co-authors did not read the manuscript thoroughly before submitting.

I have stopped the review here. If authors are willing to resubmit, please go through the whole manuscript and correct it, so it is ready for review. 

L367: Why is there a link in the text? The journal has a policy of how references should be placed in the text. They should be numbered. Please change this throughout the manuscript.

Comments on the Quality of English Language

English must be improved.

Author Response

We present here a point-by-point response to Reviewers’ comments on the manuscript: “Safeguarding Bee Health: Insights from a Collaborative Monitoring and Prevention Project against Pesticide Poisonings”, submitted to be published on Animals (animals-3316322). The relevant revisions are highlighted in yellow in the text of the manuscript.

Reviewer #2: The authors present a study on pesticide analysis in pollen. The study sound interesting. However, from reading the manuscript it becomes obvious that the manuscript was submitted unfinished. There are many technical errors. In addition, I feel that the manuscript was not originally written in English and that the translator was not of scientific background. It should not be published in the present form. The manuscript even has instruction from guidelines left within the text. It requires substantial revisions if it is to be reconsidered for publication.

We wish to thank the reviewer for his/her constructive comments. We realized the incompleteness of the first draft of the manuscript, and we tried to improve it following the reviewer's suggestions.

Some specific comments:

L20: More accepted term is “colony collapse disorder” CCD.

Done. Throughout the whole text, the original expression "Hive depopulation syndrome" was replaced by “Colony Collapse Disorder-(CCD)” as a more accepted and scientifically sounding term

L33: and is not needed?

Deleted.

L34: Replace “advanced chromatography techniques” with specific method(s) used.

The methods were specified as follows: “gas and liquid chromatography tandem quadrupole mass spectrometry (GC-MS/MS and LC-MS/MS)” (Lines 42-43).

L37: If it happened only once, particularly is not needed.

Deleted.

L38: clinical and subclinical? Do you mean lethal and sublethal? Please use widely accepted terminology of lethal and sublethal throughout the manuscript. 

As also suggested by Reviewer #1, we used the widely accepted terminology of lethal and sublethal throughout the manuscript.

L81-89: Specifics on the funding of the project go to the acknowledgment or funding section, not in the scientific part of the manuscript. Please describe the study not the project. This section should be the hypostases and/or aim of the presented study.

We have moved the following sentence to the "Acknowledgments" section (Lines 336-339): “The Authors thank the management of the Agenzia per la Tutela della Salute (ATS) of Brianza, Italy, for their support to this study as part of the activities foreseen by ”Project on Monitoring and Prevention of Bee Poisonings caused by Pesticides and Agrochemicals" as established by Resolution no. 636/2021”. We have also revised the last part of the introduction to better illustrate the purpose of the study, as suggested (Lines 86-95).

2.1 Monitoring stations. This whole section must be completely rewritten. Did you mean monitoring locations? What are sentinel stations? What are Plain Stations? Reading the description, it feels that these stations have the same characteristics. Station 1 designated as agricultural has significant industrial settlements and traffic. Then the same is for station 2. What does “Significant provincial, state, and municipal infrastructure” refer to?

We completely revised this Section 2.1, better specifying the terminology used and highlighting the differences between the areas where the four sentinel apiaries (referred to as S1 to S4) were located. We also added a map in the new Figure 1, showing their geographical location within the study area (Lines 99-120).

L104: others? Please finish the sentence

This sentence was reformulated (Line 107).

L142: What is extraordinary monitoring?

This paragraph was deleted.

A map of the locations would prove very useful.

We added a map (now called Figure 1) which shows the location of the four sentinel apiaries (Lines 118-120)

L147: Exact model to be specified?

We deleted this phrase (that was in round brackets) since the removable drawer closure could not be associated with a specific model (Line 122).

L148-151: From this sentence it can be concluded that the mortality was not measured at the sampling station but at hives adjacent to the station?

This sentence was deleted.

L158: Subtitle should be in a new line.

Done.

L166-176: How much of the supernatant after extraction procedure did you use for clean-up? With how much of the clean extract did you end up with? Did you have a concentration step? Did you follow literature protocol? If yes, please cite the work.

In the text we specified that: a) the exact volume of the supernatant was 6 mL (Line 143), b) no subsequent sample extract concentration step was necessary neither solvent change for instrumental analysis (Lines 145-146), c) each pollen sample purified extract was directly analyzed by GC-MS/MS and LC-MS/MS techniques (Line 147), and d) the method was validated according to EU DG SANTE/11312/2021 document, and enabled the assessment of potential correlations. Pesticide accumulation in pollen and all samples were analyzed in accordance to the criteria required in the SANTE document was added to the reference list and numbered as [9] (Lines 181-184).

L207-209: How did you achieve matrix matched solutions?

In the text we specified the procedure that was followed to build matrix-matched calibration curves for pesticide quantification (Lines 175-181).

L215-217: It is biggening to feel that co-authors did not read the manuscript thoroughly before submitting.

We sincerely apologize for having overlooked removing these instructions. Now they have been deleted.

I have stopped the review here. If authors are willing to resubmit, please go through the whole manuscript and correct it, so it is ready for review.

We made several revisions to the manuscript based on Editor’s and Reviewers’ comments.

L367: Why is there a link in the text? The journal has a policy of how references should be placed in the text. They should be numbered. Please change this throughout the manuscript.

As also suggested by the Editor, we numbered all the References in order of appearance in the text throughout the manuscript, and listed accordingly in the corrisponding Section.

Comments on the Quality of English Language

English must be improved.

We revised the English language throughout the manuscript.

Round 2

Reviewer 1 Report

Comments and Suggestions for Authors

Thank you for addressing the points mentioned in the first review. I don't have any further comments. 

Author Response

We present here a point-by-point response to the second round of reviewers’ comments on the manuscript: “Safeguarding Bee Health: Insights from a Collaborative Monitoring and Prevention Project against Pesticide Poisonings”, submitted to be published on Animals (animals-3316322). The relevant revisions to the previous version of the manuscript (provided following the first round of the review process) are highlighted in yellow in the text of the new version. Furthermore, the ORCID identifier has been added to the corresponding Author (Line 16) as required by the policies of the University of Milan which will partially support APC expenses; in this regard, the relevant reference has also been added in the Acknowledgments section (Lines 354-355). Besides, the reference number 9 regarding the original version from which Figure 1 was derived has been mentionend in the Figure 1 legend and has also been added to the References section. Consequently, the sequence of references in the text and in the corresponding section has been renumbered. Finally, the reference number 25, which was missing from the References section due to a typographical error despite being cited in the text (formerly as number 24), has been added.

REVIEWER #1

Thank you for addressing the points mentioned in the first review. I don't have any further comments.

Dear Reviewer#1, on behalf of all the co-authors I thank you for your valuable comments which allowed us to improve our manuscript. Kind regards, Michele Mortarino

Reviewer 2 Report

Comments and Suggestions for Authors

The authors have now corrected the manuscript that was earlier rejected due to it being incomplete. There are, however, still some issues that need to be addressed. I really like the discussion section, as it nicely explains the results found during the study, although some minor changes to this section are also needed.  

Specific comments:

L31: Frequent limitations, or frequently limiting factors, however I would suggest revising the whole sentence to make it more understandable.

L43: instead of “specific” use “a number of”

L77: In this regard…

L78: need to be overcome or addressed

L81: coma before d) and watch for spacing

L92-95: This is the introduction section. It should not contain any results and conclusion drown from the results of the study. This belongs in the conclusions part of the manuscript.

L99-118: It appears that all of the sampling sites were with high-traffic roads compared to one another. It should be stated which site has the highest traffic and which the lowest. It would be useful to also add the names of the villages or GPS coordinates of the hives.

L111: The hills are between 230-1400 meters? At which altitude was the apiary stationed? Did the bees form this apiary fly all the way from 230-1400 meters?

L115L The same as previous, but here the elevation is even greater. I find it very difficult to believe that the bees were flying from 600-2400 meters elevation.

L126-127: You state that 20 pollen samples were collected from each apiary during the study. Each apiary has 3 hives. Were the samples from the three hives at each apiary mixed together? From the stated I would suspect that they were, however analyzing them separately would give much better statistics. Please specify. 

L186: What do you mean by molecules or groups of analytes? Do you refer to pesticides?

L188-191: Why do you specifically mention non-polar phytopharmaceuticals in this paragraph? As presented now it suggests that you looked at 238 compounds and then the non-polar phytopharmaceuticals in addition to these 238. Please separate them in two paragraphs.

L208-209: I do not think that it is necessary to have “(data not shown)”. If you make a statement that there was no mortality, then that should be considered relevant data.

L240 and L276 and L294: What is “principles”? It should maybe be changed to compound?

L246: What are mineral matrices?

L249-253: This sentence needs to be rewritten.

L276-279: “…primarily as a pre-emergence herbicide, mainly 278 in post-emergence applications for maize…” Please clarify, is it supposed to be used as pre-emergency but is instead used as post-emergency?

Author Response

We present here a point-by-point response to the second round of reviewers’ comments on the manuscript: “Safeguarding Bee Health: Insights from a Collaborative Monitoring and Prevention Project against Pesticide Poisonings”, submitted to be published on Animals (animals-3316322). The relevant revisions to the previous version of the manuscript (provided following the first round of the review process) are highlighted in yellow in the text of the new version. Furthermore, the ORCID identifier has been added to the corresponding Author (Line 16) as required by the policies of the University of Milan which will partially support APC expenses; in this regard, the relevant reference has also been added in the Acknowledgments section (Lines 354-355). Besides, the reference number 9 regarding the original version from which Figure 1 was derived has been mentionend in the Figure 1 legend and has also been added to the References section. Consequently, the sequence of references in the text and in the corresponding section has been renumbered. Finally, the reference number 25, which was missing from the References section due to a typographical error despite being cited in the text (formerly as number 24), has been added.

REVIEWER #2

Comments and Suggestions for Authors

The authors have now corrected the manuscript that was earlier rejected due to it being incomplete. There are, however, still some issues that need to be addressed. I really like the discussion section, as it nicely explains the results found during the study, although some minor changes to this section are also needed. 

We wish to thank once again the reviewer for his/her constructive comments and his/her suggestions to further improve the manuscript.

Specific comments:

L31: Frequent limitations, or frequently limiting factors, however I would suggest revising the whole sentence to make it more understandable.

We revised the sentence as follows: “Poor knowledge about the level of exposure of bee colonies to pesticides and agrochemicals, whether from the environment or beekeeping management practices, is a major limiting factor in preventing these diseases” (Lines 31-33).

L43: instead of “specific” use “a number of”

Done as suggested.

L77: In this regard…

Done as suggested.

L78: need to be overcome or addressed

Done as suggested.

L81: coma before d) and watch for spacing

Done as suggested.

L92-95: This is the introduction section. It should not contain any results and conclusion drown from the results of the study. This belongs in the conclusions part of the manuscript.

As suggested, this sentence has been moved to the Conclusions section (Lines 319-322).

L99-118: It appears that all of the sampling sites were with high-traffic roads compared to one another. It should be stated which site has the highest traffic and which the lowest. It would be useful to also add the names of the villages or GPS coordinates of the hives.

The manuscript now states that the area within the potential bee flight range around S1 was characterized by the highest vehicular traffic density among all the sampling sites, being near the intersection of two highways traveled daily by tens of thousands of vehicles (Lines 104-107). Similarly, the manuscript now specifies that the area within the potential bee flight range for the S4 sampling site is characterized by the lowest vehicular traffic intensity compared to all others (Lines 121-122). Finally, the text now specifies the locations of the sentinel apiaries, in the municipalities of Agrate (S1), Monza (S2), Valmadrera (S3), and Crandola Valsassina (S4) respectively (Lines 100-102).

L111: The hills are between 230-1400 meters? At which altitude was the apiary stationed? Did the bees form this apiary fly all the way from 230-1400 meters?

The manuscript now states that S3 was located at 450 meters above sea level, with potential bee flight range covering a hilly area of altitude ranging from 230 to about 1400 meters above sea level (Lines 112-113).

L115L The same as previous, but here the elevation is even greater. I find it very difficult to believe that the bees were flying from 600-2400 meters elevation.

For both S3 and S4 sentinel apiaries, the indicated altitude bands represented areas where bees were likely to forage. Within these bands, summer blooms of pastures and spontaneous flora, such as rhododendron and dandelion, attracted bees and were potentially accessible, although it was unlikely at altitudes above 2,000 meters. Accordingly, the manuscript now states that S4 was located at 850 meters above sea level, with potential bee flight range covering a hilly area of altitude ranging from 600 to about 2000 meters above sea level (Lines 118-119).

L126-127: You state that 20 pollen samples were collected from each apiary during the study. Each apiary has 3 hives. Were the samples from the three hives at each apiary mixed together? From the stated I would suspect that they were, however analyzing them separately would give much better statistics. Please specify.

For greater clarity, we have added the following sentence: “During each collection, samples were prepared by combining pollen from the three hives of each sentinel apiary. This approach ensured the collection of sufficient pollen quantities during the entire sampling period, even at times when the harvest was scarce (Lines 133-136).

L186: What do you mean by molecules or groups of analytes? Do you refer to pesticides?

The term 'pesticide' is now included in the text (Line 196).

L188-191: Why do you specifically mention non-polar phytopharmaceuticals in this paragraph? As presented now it suggests that you looked at 238 compounds and then the non-polar phytopharmaceuticals in addition to these 238. Please separate them in two paragraphs.

Indeed, we believe that the term 'non-polar' referring to the chemical nature of the compounds analyzed in the study, is here misleading. Consequently, this specification has been removed.

L208-209: I do not think that it is necessary to have “(data not shown)”. If you make a statement that there was no mortality, then that should be considered relevant data.

In accordance with this suggestion, the parenthetical expression '(data not shown)' has been removed.

L240 and L276 and L294: What is “principles”? It should maybe be changed to compound?

As suggested, the term 'principle(s)' has been replaced with 'compound(s)' (Line 250-286-304, respectively).

L246: What are mineral matrices?

The term 'mineral' has been replaced with 'inorganic'.

L249-253: This sentence needs to be rewritten.

The sentence has been rewritten as follows: “This primarily concerns pollen, which is used by bee nurses for larval feeding, and its various sublethal and chronic pathogenic effects. These may lead to a significant decrease in queen fertility, as well as a reduction in bee longevity, immune function, orientation, and social behavior. Moreover, these effects may contribute to CCD, often in combination with other factors [14].(Lines 259-264).

L276-279: “…primarily as a pre-emergence herbicide, mainly 278 in post-emergence applications for maize…” Please clarify, is it supposed to be used as pre-emergency but is instead used as post-emergency?

It has now been clarified in the text that the herbicide Pendymetalin is primarily used as a pre-emergence herbicide on spring crops (Lines 288-289).
